# Stem Cell Therapies for Epidermolysis Bullosa Treatment

**DOI:** 10.3390/bioengineering10040422

**Published:** 2023-03-27

**Authors:** Argyrw Niti, Georgios Koliakos, Anna Michopoulou

**Affiliations:** 1Biohellenika Biotechnology Company, Leoforos Georgikis Scholis 65, GR 555 35 Thessaloniki, Greece; 2Laboratory of Biological Chemistry, Medical School, Aristotle University of Thessaloniki, GR 541 24 Thessaloniki, Greece

**Keywords:** epidermolysis bullosa, stem cells, ex vivo gene therapy

## Abstract

Epidermolysis bullosa (EB) includes a group of rare skin diseases characterized by skin fragility with bullous formation in the skin, in response to minor mechanical injury, as well as varying degrees of involvement of the mucous membranes of the internal organs. EB is classified into simplex, junctional, dystrophic and mixed. The impact of the disease on patients is both physical and psychological, with the result that their quality of life is constantly affected. Unfortunately, there are still no approved treatments available to confront the disease, and treatment focuses on improving the symptoms with topical treatments to avoid complications and other infections. Stem cells are undifferentiated cells capable of producing, maintaining and replacing terminally differentiated cells and tissues. Stem cells can be isolated from embryonic or adult tissues, including skin, but are also produced by genetic reprogramming of differentiated cells. Preclinical and clinical research has recently greatly improved stem cell therapy, making it a promising treatment option for various diseases in which current medical treatments fail to cure, prevent progression, or alleviate symptoms. So far, stem cells from different sources, mainly hematopoietic and mesenchymal, autologous or heterologous have been used for the treatment of the most severe forms of the disease each one of them with some beneficial effects. However, the mechanisms through which stem cells exert their beneficial role are still unknown or incompletely understood and most importantly further research is required to evaluate the effectiveness and safety of these treatments. The transplantation of skin grafts to patients produced by gene-corrected autologous epidermal stem cells has been proved to be rather successful for the treatment of skin lesions in the long term in a limited number of patients. Nevertheless, these treatments do not address the internal epithelia-related complications manifested in patients with more severe forms.

## 1. Introduction

The skin is the largest organ of the body constituting the barrier between the organism and the external environment and protecting it against physical, chemical and biological assailants, dehydration, UV radiation, etc. [1]. Human skin consists of a stratified epithelium externally, named the epidermis, and the underlying connective tissue, named the dermis, mainly resided by fibroblasts [2]. The functions and homeostasis of skin require stable organization and cohesion between the epidermis and the dermis. These tissue layers are interconnected via the dermo-epidermal junction (DEJ), which comprises the keratinocytes of the basal epidermal layer, the specialized dermo-epidermal extracellular matrix (ECM) named the basement membrane and the uppermost dermis (papillary) [3].

The DEJ is a highly specialized ECM structure, which promotes strong epidermal/dermal attachment and epithelial cell polarization [4]. The basal lamina of epithelia is characterized at the ultrastructural level by the presence of regularly distributed structures, the anchoring complexes. These structures consist of basal membrane electron-dense thickenings, the hemidesmosomes (HDs), the anchoring filaments, which span the lamina lucida and insert into the lamina densa and the anchoring fibrils which originate in the lamina densa and project into the upper region of the papillary dermis [5]. The molecular organization of HDs comprises three classes of proteins: the cytoplasmic plaque proteins acting as linkers for elements of the cell cytoskeleton (intermediate filaments composed by keratins), the transmembrane cell receptors connecting the cell interior to the basement membrane associated proteins [6]. The main components of the cytoplasmic plaque are the bullous pemphigoid antigen 230 (BP230) [7], plectin and other proteins named HD-1, IFAP300 (Intermediate Filament-Associated Protein) and P200. BP230 and plectin are implicated in the organization of architectural structures [6]. The transmembrane components include integrin α6β4 and BP180 (bullous pemphigoid antigen 2 or type XVII collagen). Integrins α6β4 connect intracellularly with the intermediate filaments network (keratins) and extracellularly with laminin 332, while BP180 plays a role in epithelial-mesenchymal adhesion. Hence, laminin 332 serves as a bridge between α6β4 and the dermal matrix and constitutes an irreplaceable component of this supramolecular network, crucial for the formation of HDs and the maintenance of stable adhesion [8,9].

Epidermolysis bullosa (EB) constitutes a family of genodermatopathies characterized by the formation of blisters as a result of structural skin fragility and other epithelial tissues in response to minor mechanical pressure. Clinical manifestations vary from localized to generalized blisters and lesions of the skin throughout the body and oral cavity related to secondary deficits, including chronic wounds, scars, deformities, infections and complications to multiple internal organs. The skin is the main organ affected but other epithelial tissues are often implicated including the external eye, upper respiratory tract, genitourinary tract, and gastrointestinal tract [10]. In conclusion, EB is a multifactorial and multisystemic disease that greatly affects the quality of life that is characterized by significant mortality and morbidity rates. The various subtypes of EB result from mutations in the genes encoding many different proteins, which are indispensable for maintaining the structural stability and strong adhesion of the keratinocytes in the underlying basement membrane and dermal tissue (Figure 1). Diagnosis and categorization are based on the collective findings obtained through a detailed personal and family history, combined with the results of immunomapping, transmission electron microscopy, and in some cases, by searching for the defective gene with molecular techniques. In recent years, the focus has been driven on the elucidation of the clinical and molecular basis of the disease. The progress made in translational research has initiated innovative clinical trials of new cell- and gene-based therapies [11].

## 2. Types of Epidermolysis Bullosa

The term epidermolysis bullosa (EB) was first described in 1886 [12], while in 1962 the first satisfactory classification system was proposed by Pearson [13]. Today it is classified into four main types: (a) simplex (SEB), (b) junctional (JEB), (c) dystrophic (DEB) and (d) mixed (Kindler syndrome), mainly based on the level of separation within the skin and the following blister formation (Figure 1). The spectrum of EB spans more than 32 clinical subtypes [10] with pathogenic mutations in at least 18 distinct genes [14], depending on the mode of inheritance, clinical findings, and associated molecular and genetic disorders.

### 2.1. Epidermolysis Bullosa Simplex (EBS)

It is the type with the highest prevalence and a wide range of skin findings among the many subtypes of EBS, but also the mildest form of the disease, although the bullae are painful. Blisters appear on the outer layer of the skin, within the epidermis (Figure 1), and according to the distribution, frequency and severity of blisters the following variants have been described: (1) “localized” EBS with blisters limited, (2) “generalized” EBS with blisters extended all over the body and (3) Dowling-Meara (EBS-DM), where the vesicles are also generalized but show a distinct “herpetic” or clustered pattern [15]. For its diagnosis, immunofluorescence and transmission electron microscopy are used, while molecular genetic testing detects mutations in the intermediate cytoskeleton proteins keratins 5/14 (KRT5/KRT14), EXPH5 (Exophilin-5), TGM5 (Transglutaminase-5) genes [16]. There are other forms of EBS, less common, such as EBS with optic pigmentation (EBS-MP) characterized by abnormalities in skin pigmentation, EBS with muscular dystrophy (EBS-MD) accompanied by muscle weakness of the extremities, and acantholytic EBS with alopecia and nail loss [17].

### 2.2. Junctional Epidermolysis Bullosa (JEB)

JEB is characterized by a disruption of the dermal-epidermal attachment at the level of the lamina lucida and by defects in HDs (Figure 1). Mutations affecting either laminin 332 or integrin α6β4 can cause the disease, which affects the skin and mucous membranes. In the skin, laminin 332 is synthesized by keratinocytes and it is an epithelial-basement membrane specific variant [18]. JEB is generally subdivided into three subtypes named: (1) Herlitz (lethal), (2) non-Herlitz (non-lethal) and (3) JEB with pyloric atresia. All subtypes are inherited in an autosomal recessive manner. Herlitz JEB is characterized by the complete absence of laminin 332, non-Herlitz is caused by missense mutations that result in a reduction of functional laminin 332 or by the complete absence of collagen XVII and JEB with pyloric atresia is caused by mutations in the genes encoding for either α6 or β4 integrin subunits [19]. Finally, LOC syndrome (Laryngeal-onycho-cutaneous syndrome) is a subtype of JEB caused by specific mutations in the LAMA3 gene [20]. It is characterized by localized blisters and erosions, particularly on the face and neck, combined with the effects of upper airway disease, nail abnormalities, erosions, granulation tissue and enamel hypoplasia.

### 2.3. Dystrophic Epidermolysis Bullosa (DEB)

In DEB, the level of disruption that results in the blisters is located below the lamina densa at the level of anchoring fibrils (Figure 1) [21]. It is mainly caused by mutations in the COLVII-A1 (Collagen-VII-A1) gene that encodes type VII collagen (COLVII), an important component of the anchoring fibrils [22]. In the skin, COLVII is produced and deposited by epidermal keratinocytes and papillary dermal fibroblasts [23]. Dystrophic EB (DEB) is divided into two main types based on the mode of transmission, autosomal dominant (DDEB) and autosomal recessive (RDEB) [24]. It can be subdivided into at least 11 clinical subtypes, which also take into account the phenotypic characteristics involved [25].

DDEB is characterized by reduced type COLVII expression and it is considered to have a good prognosis. Usually, the blisters are mild and limited to areas such as the hands, feet, knees, elbows and esophagus. Scarring and marks from previous wounds often result in contracture of the joints and membranes of the mouth, dystrophic nails or loss of nails, and esophageal stricture [26]. Autosomal recessive RDEB is the most severe of all DEB subtypes because it is caused by an absence of a significant reduction in type COLVII expression. There are three main subtypes of RDEB—severe generalized RDEB (formerly called Hallopeau-Siemens RDEB), non-Hallopeau-Siemens RDEB or generalized intermediate, and reverse RDEB. The most severe subtype is a generalized form of RDEB. Clinical findings include generalized blisters at birth leading to extensive scarring, hypo- or hyperpigmentation, corneal blisters, growth retardation, multifactorial anemia, esophageal strictures, and deformities of the hands and feet [27]. Patients have a low survival rate and an expectancy of life of less than 30 years, due to severe renal complications or aggressive squamous cell carcinoma [28]. Generalized, intermediate RDEB is characterized by less severe symptoms and with lower skin involvement and a risk of secondary complications. Moreover, these patients still have a significant risk of developing squamous cell carcinoma even if the risk of metastases and death is lower than that in patients with severe generalized RDEB [29].

### 2.4. Mixed Epidermolysis Bullosa—Kindler Syndrome

Kindler syndrome is characterized by generalized blisters at birth that may form at multiple levels of the basement membrane or beneath the epidermal layers and later development of photosensitivity. Complications encountered include atrophic scars, nail dystrophy, severe colitis, esophagitis, and urethral stricture. Mutations in the FERMT1 (Four point one Ezrin Radixin and Moesin − 1) gene are the main cause of Kindler syndrome, which lead to premature termination of translation and loss of the kindlin-1 protein [30].

## 3. Introduction to Stem Cells

Stem cells have long been thought of as an evolution and revolution in modern biomedical science and have held much hope and promise due to their potential for regeneration. Many therapeutic applications of stem cells in humans are known today: from bone marrow transplantation to skin repair. Specifically, the characteristics that a cell must have in order to be considered a stem cell are (i) the ability to divide and self-renew for a long time through cell divisions, (ii) to be undifferentiated (clonality) and (iii) to be able to differentiate into others cell types [31]. Due to their important regenerative abilities, stem cells have increased hope considering the treatment of a significant number of diseases and are being evaluated in numerous clinical trials targeting different conditions. However, much work remains to be done at the laboratory and clinical level to understand how to use these cells in cell therapies to treat disease, also referred to as regenerative or regenerative medicine.

Stem cells vary widely depending on their origin, ability to proliferate, and the range of mature cell types they can produce. The classification by the extent to which they can differentiate into different cell types includes the following main groups (Table 1):Totipotent Cells: They have the ability to differentiate into all possible cell types. For example, a fertilized oocyte and the first few cells that result from the division of the zygote are totipotent cells [32].Pluripotent Cells: They have the ability to differentiate into cells that come from the 3 germ layers—ectoderm, endoderm, and mesoderm—from which all tissues and organs derive [33]. Examples include embryonic stem cells (ESCs), as well as induced pluripotent stem cells (iPSCs) which arise from reprogramming somatic cells [34].Multipotent Cells: They have the ability to differentiate into cells from a single germ layer [35]. Examples include mesenchymal stem cells (MSCs) which can differentiate into mesoderm-derived tissue such as adipose tissue [36] or ectoderm tissue such as neuronal tissue [37].Oligopotent Cells: They have the ability to differentiate into a few lineages within a single tissue. This category includes hematopoietic stem cells (HSCs) which can differentiate into lymphoid or myeloid stem cells [38].Unipotent Cells: They have the ability to only produce cells of their own type, such as muscle stem cells which can differentiate into only mature muscle cells [39].As for the classification based on their origin, stem cells can be classified as below:Adult Stem Cells: they come from the “niches” in adult tissues. They are characterized as multipotent because they can self-renew, but have a low ability to differentiate. They can mature in cells of the tissue or organ from which they originate but also play a key role in their maintenance and repair [40].Advanced Perinatal embryonic stem cells: Perinatal stem cells (e.g., from aborted embryos or umbilical cord blood). They are the various types of stem cells found in the umbilical cord unit, such as hematopoietic stem cells (HSCs), mesenchymal stem cells (MSCs), amniotic epithelial cells (AMSCs), chorionic mesenchymal stem cells (CMSCs) and progenitor endothelial stem cells, as well as earlier cell types such as unrestricted somatic stem cells (USSCs) and very small embryonic-like cells (Very Small Embryonic Like Stem Cells, VSELs). The majority of the scientific community characterizes them as pluripotent and not simply as multipotent cells, which gives them the characteristic of the so-called plasticity of these stem cells and has led to a wide range of clinical applications [41].Embryonic stem cells: Embryonic cells that arise during the first four days after fertilization are considered totipotent and form all the cells of the embryo as well as extraembryonic formations, such as the placenta. Embryonic stem cells that arise after four to five days are considered pluripotent, they come from inside the blastocyst and can transform into all cell types of an organism [42]. A third class of “embryonic” cells, so-called induced pluripotent cells (iPSCs), has been added in recent years. iPSCs are developed through the genetic manipulation of differentiated cells [43].

## 4. Cellular Therapy Options for EB

Despite the progress in the field, there are still no established effective causative treatments for EB. The management of EB is primarily symptomatic and aims to prevent relapses [44]. The basic principles of care for all patients with EB are the avoidance of blisters, a protective covering of the skin, and the prevention of complications and infections, with careful wound care, facilitated by the use of sterile synthetic, non-adhesive bandages. Nevertheless, in the last decade, the rapid scientific discoveries in gene and cell therapies paved the way for new treatments and some of them have been approved for clinical trials (Table 2). Despite the fact that keratinocytes represent the main cell target in EB, the discovery of the disease’s systemic nature led to the investigation of alternative cellular sources for cell therapies including HSCs and MSCs [45]. Among the most investigated cell therapies for EB are bone more transplantation (BMT) and cell therapies with allogeneic MSCs, allogenic fibroblasts and clinical use of iPSCs [44,45]. BM–derived cells and MSCs have certain mechanisms of action that make them an attractive candidate for EB treatment. Both of them are able to migrate to damaged tissue sites where they can induce tissue regeneration and/or reduction of fibrosis through paracrine activities [46]. Moreover, if we consider EB as systemic inflammatory diseases rather than skin limited-disorders then, BMT and MSCs transplantation may have a beneficial outcome as a treatment through the immune-modulatory properties of these cells [47]. Additionally to that, it was shown that a sub-population of stem cells from the BM are known to be able to differentiate along the epithelial lineage (potential epithelial progenitors) and engraft the skin and therefore, they can be used as cellular sources of the missing or altered ECM protein, such as COLVII in RDEB [46,48]. A small number of patients have received so far genetically “corrected” keratinocytes or fibroblasts as a treatment for EB. These so-called gene therapies consist of ex vivo viral transfer of a functional cDNA copy of the defective endogenous gene in the above-mentioned cell types which are isolated from the patient’s skin biopsies, expanded in vitro and grafted back to the skin [49].

### 4.1. Hematopoietic Stem Cells (HSCs) Transplantation

As mentioned above, the generalized severe forms of EB, such as RDEB are extended to other body sites besides the skin and are life-threatening. The development of allogeneic BMT protocols for skin disorders was based on the observation that a fraction of BM cells is able to migrate to the wound site and enhance the wound healing process [45]. So far, BMT therapies have been investigated for RDEB and JEB treatment, but they have only shown some transient beneficial clinical outcomes in a small number of RDEB patients [49]. The BM primarily contains two different types of stem cells, i.e., HSCs and non-HSCs, including MSCs. Only a few published studies exist to assess the role of allogeneic HSC transplantation for RDEB treatment [50,60].

The ability of BM cells to migrate to wounded areas facilitating the healing process was first demonstrated in mice [61]. In this study, green fluorescent protein (GFP) labeled BM-derived cells detected in the dermis, epidermis and cutaneous appendages after BMT enhanced the wound healing process [61]. The transplantation of allogenic BM-derived cells into mice showed that HSCs resulted in the healing of the blisters, expression of COLVII and formation of anchoring fibrils in contrast to non- HSCs which did not give satisfactory results [62]. Subsequently, a study of BMT in mice showed that a specific subpopulation of BM-stem cells (Lin−/PDGFRα+) have the plasticity to differentiate into epithelial progenitors, contribute to the generation of new keratinocytes and increase COLVII [48]. To sum up, data from pre-clinical studies suggesting phenotypic improvement observed in COLVII -deficient mice after treatment with stem cells from BM encouraged the effort for human studies.

A potential benefit of BM stem cell therapies is likely based on the capacity of BM-derived cells to differentiate into skin cells when located in the right microenvironment [63]. BM-derived MSCs cells have been shown to differentiate into epidermal keratinocytes, follicular epithelial cells, dendritic cells, sebaceous gland cells and endothelial cells [64]. Moreover, it was suggested that the infusion of allogeneic BM stem cells can promote an increase of COLVII and reduce blistering in patients with DEB [50]. Also, other epithelial proteins involved in forms of EB associated with severe morbidity may be therapeutically affected by BM-derived stem cells [62]. It has been noticed that donor cells differentiated into keratinocytes were located in the patients of a BMT study and remained at least for three years [65]. Finally, it should be noted that BMT studies in humans or mice have shown that BM-derived cells, which are keratin-positive, can be located in skin stem cell niches, such as skin epidermis, hair follicles, and sebaceous glands [48].

Despite the benefits that patients—primarily those suffering from severe generalized RDEB—may experience after allogeneic hematopoietic and BM transplantation, these treatments have some important drawbacks. The transplantation of allogeneic BM and hematopoietic cells are often related to mortality due to immunoablative regimens that may be toxic or result in opportunistic infection and/or graft-versus-host disease [51,66]. As a matter of fact, the first clinical trial (NCT00478244, Table 2) using BMT for the treatment of seven children with severe RDEB was performed after fully immunoablative chemotherapy. Although, there was an increase of COLVII expression and donor cells in the recipient skin and decreased blister formation the potential benefit of the treatment was hampered by regimen-related toxicity [50]. In another hematopoietic stem cell transplantation (HCT) trial, two children with RDEB showed increasing donor chimerism in whole blood and skin donor chimerism, but without long duration something which improved their clinical situation, but only temporarily [67]. Later on, a clinical trial with BMT in two children with RDEB could not be completed due to their premature death caused by acute graft rejection and graft-versus-host disease [68]. Therefore, the subsequent studies performed focused on the safety of BMT against the complications recorded in graft-versus-host disease and the toxicity of the myeloablative preparation [67,68]. Worldwide many clinical trials are still under development leading to promising results of BMT in patients with RDEB and stem cell transplantation protocols are currently being modified with the aim to improve safety (NCT02582775, NCT01033552) [69].

HCT in patients with JEB did not give satisfactory results. The transplant of peripheral blood allogeneic HSCs in patients with JEB followed by myeloablative conditioning resulted in an initial improvement of the healing process, however, the patients passed away due to complications related to the rejection of the graft [70,71].

Although HCT appears to be a promising option for patients with RDEB, many concerns remain regarding the safety of the protocols applied, which has led to the use of other non-hematopoietic cell populations. An improvement of BMT was the addition of donor-derived MCSs when BMT using as a combination therapy [51].

### 4.2. Cytotherapies with Mesenchymal Stem Cells (MSCs)

The latest clinical studies using stem cells for the treatment of EB mainly focus on the use of MSCs and primarily on the treatment of RDEB (Table 2).

Among adult stem cells, MSCs are isolated from many adult tissues, they are capable of multilineage differentiation and present low immunogenicity. Moreover, MSCs demonstrate immunomodulatory properties, they can migrate to the damaged tissue and stimulate tissue regeneration and reduction of fibrosis mainly through paracrine activities [46,47]. Hence, they are an attractive option for allogeneic use in several inflammatory skin diseases including EB. BM was initially regarded as the major source of MSCs. Adipose-derived MSCs (AT-MSCs) first started to be used in regenerative medicine, as an alternative or in addition to BM-derived MSCs (BM-MSCs) [72,73]. Nowadays, several tissues are used for the isolation of MSCs every one of each displaying some advantages against the others. Those include skin, liver, bile ducts, lungs and teeth and birth-associated tissues, such as the placenta, amnion, umbilical cord (UC-MSCs) and umbilical cord blood (UCB-MSCs) [72,73].

BM-MSCs’ major advantage is the maintenance of substantial multilineage differentiation potential [74] and significant paracrine function related to angiogenesis [75]. However, BM-MSCs are isolated through an invasive procedure accompanied by pain and risk of infection and have lower proliferation capacity and lifespan as compared to other adult tissue-derived sources [72,74]. AT-MSCs are isolated from the subcutaneous tissues and can be abundantly available and harvested by liposuction or lipectomy which are much less invasive procedures. AT-MSC’s advantages as compared to BM-MSCs are proliferation rate and lifespan. Nevertheless, they contain more committed progenitor cells and lower secretion of proangiogenic molecules and cytokines [72,73]. Umbilical cord blood and umbilical cord are rich sources of highly proliferating and young MSCs with better isolation yield, similar secretion of cytokines and differentiation potentials as compared to BM-MSCs. Moreover, UCB and UC-derived MSCs are readily available from cord blood banking [46,72].

RDEB is an inherited skin disorder that results from the lack of functional COLVII and for which no effective therapy exists. Several therapeutic approaches, either by intravenous infusion or direct local administration of MSCs to chronic wounds, have been initiated based on pre-clinical data showing that intradermal injections of human MSCs showed a dose-dependent, significant high production and local deposition of COLVII associated with the restoration of connective fibrils and superior dermal-epidermal integrity [76]. Conget et al. 2010 described COLVII replenishment at the dermal-epidermal junction upon intradermal injection of healthy donor MSCs into chronic wounds of two patients with RDEB [77]. Nevertheless, intradermal administration of MSCs has limitations that are incapable of addressing mucosal and other systemic complications [78]. In addition to that, later clinical studies were developed, which questioned the impact of intravenously/systemically administered MSCs on the restoration of COLVII and anchoring fibrils in patients with RDEB (see Table 2) (NCT04520022, NCT02323789, 2012-00894-87) [46,52,53]. In general, intradermal or intravenous injections of MSCs have shown some clinical benefits for RDEB patients. As shown in Table 2, these benefits include mainly a reduction in disease activity, pain and itch and an increase in the quality of life. A new population of dermal MSCs has quite recently been characterized as ATP-binding Cassette Transporter, Subfamily B, Member 5 positive MSCs (ABCB5+ MSCs) and capable of exerting therapeutic immunomodulatory functions [79]. Data from the application of these cells for the treatment of chronic wounds have shown that ABCB5+-MSCs, likely exert their beneficial activity by shifting the M1 proinflammatory macrophages to anti-inflammatory, repair-promoting, M2 macrophages [80]. Moreover, there is evidence suggesting that ABCB5+-MSCs present superior homing potential to injured tissues than BM-MSCs and are capable of secreting COLVII. Results from a phase I/II clinical study (NCT03529877) conducted recently have shown a beneficial effect of the administration of ABCB5+-MSCs in patients with RDEB. However, as mentioned this trial is quite recent and also the potential deposition of COLVII in skin and mucosa has not been assessed [47].

Despite the encouraging results, improved application techniques and optimization of dosing protocols and frequency of administration of allogeneic MSCs are still required and must be evaluated in future clinical trials. Moreover, it may be important that the treatment starts at a young age to prevent the onset of serious functional damage.

### 4.3. Genetically Corrected Autologous Epidermal Stem Cell and Fibroblasts Therapies

Ex vivo gene modification therapies for EB involve the correction of related mutant genes that are delivered using retro- or lentiviral vectors in patients’ epidermal stem cells or fibroblasts in vitro, and then engraftment or injection of the corrected cells back to the patient [81]. However, the genetil modification of epidermal keratinocytes was quite challenging generating a number of issues to be addressed including 1. Transduction efficiency, 2. Maintenance of the transgene expression in the long term, 3. Immunogenicity induced by the transgene and 4. implantation of the transduced cells [82].

Previous advances in epidermal stem cell cultures allowed the generation of entire epidermal sheets that were initially used to treat patients with severe burns [83,84] The idea of using ex vivo modified autologous epidermal stem cells to produce epidermal sheets in vitro for transplantation of patients with their own “corrected” epidermis was a revolutionary idea for treating. Despite the challenges, the first ex-vivo gene therapy for JEB came up in 2006 involving the preparation and engraftment of epidermal sheets produced by the patient’s autologous corrected ESCs for the LAMB3 gene (Laminin Subunit Beta 3) [85]. Follow-up studies showed continued clinical benefits of the patient and expression of laminin 332 in the DEJ for more than six and a half years and absence of serious adverse effects (NCT05111600) [54]. Successful ex vivo gene therapies for JEB were later performed in two more patients with mutated LAMB3 gene [86,87]. While the first two attempts were limited to engraftment of small skin areas for safety reasons the last patient’s life-threatening condition led to engraftment of 80% of the skin are and the outcome was life-saving and lasted in the long term. These therapies are likely safe and efficient in the long term according to the results of a 16-year follow-up.

The success of delivering the corrected LAMB3 gene after ex vivo epidermal cell correction in the form of an epidermal sheet in patients with JEB was attributed to a number of parameters. Firstly, holoclone epidermal stem cells with the transgene were targeted in the grafted skin of the patients. Holoclones have the capacity to renew the epidermis several times which most likely provided those treatments with durability [88]. This event may have been due to the survival and replicative advantages of cells after re-introduction of functional laminin 332. As shown before, laminin 332 plays a key role in mediating keratinocytes adhesion, while deprived of laminin 332 in vitro they detach from the culture surface [9]. Therefore, re-introducing laminin-332 must have given a survival advantage to the corrected cells in vitro and in vivo over the cells lacking functional laminin 332. Last but not least, none of the patients manifested immune reactions to the gene product. A possible explanation for that was the selection of patients with missense mutations rather than patients carrying full mutations. The fact that these patients expressed some even defective laminin 332 means that there was a chance for those patients’ immune systems to have developed some tolerance towards the laminin β3 chain [88].

The favorable outcomes observed after LAMB3 replacement therapy encouraged the introduction of similar trials to evaluate the safety and efficacy of autologous epidermal skin grafts expressing COLVII-A1 (NCT02984085) and collagen XVII (NCT03490331) in RDEB and JEB, respectively (Table 2) [89]. However, there is one significant obstacle in gene therapy for RDEB, which is the significantly greater size of the COLVII-A1 transcript [88]. Overcoming this problem generated the first successful COLVII-A1 gene therapy trial in seven patients (NCT01263379) [59]. The treatment resulted in improved wound healing and proper localization of COLVII in the basement membrane zone, whereas no serious adverse effects were reported. Nevertheless, transgene expression decreased over time, and wild type protein expression was maintained only in two patients (NCT01263379) [57,58,59]. In an attempt to address the problem of COLVII optimal assembly and maintenance in normal anchoring fibrils the GENEGRAFT project was generated. Evidence suggesting that COLVII produced from keratinocytes and fibroblasts is necessary for appropriate assembly into anchoring fibrils led to the production and engraftment of a full-thickness skin equivalent by introducing the transgene in both keratinocytes and fibroblasts (NCT04186650, EBGraft, Table 2).

Last but not least, fibroblasts can be another source of COLVII and can be easily isolated and expanded extensively in vitro. Therefore, fibroblasts could also be a potential candidate for ex vivo gene therapy. Indeed, two clinical trials have been developed so far (NCT02493816, NCT02810951, Table 2) evaluating the safety and efficacy of intradermal injection of gene modified autologous fibroblasts for RDEB [55,56]. One trial revealed functional COLVII expression and new anchoring fibrils formation after the intradermal injections of corrected autologous fibroblasts [56]. The other one showed increased expression of COLVII at the site of injection for 12 months but no functional anchoring fibrils formation [55]. Either way, it should be noted that therapies with intradermal injection of corrected autologous fibroblasts cannot provide a therapeutic solution in the long-term and multiple applications are required to maintain the therapeutic effect.

### 4.4. Induced Pluripotent Stem Cells (iPSs)

iPSCs are produced from somatic cells that are re-programmed to acquire the capacity for pluripotent differentiation and unlimited self-renewal. The ability to create pluripotent cell lines can be used to model human diseases [34]. Especially in RDEB, where access to patients’ stem cells is not always easy, the production of iPSCs increases the opportunity for autologous stem cell transplantation. As proof of this, transplantations of differentiated skin and bone marrow cells from iPSCs in stem-matched mouse models seem to have no important immune response [90].

Keratinocytes and fibroblasts have been produced several times from iPSCs derived from patients with different subtypes of DEB [91,92,93,94,95,96]. iPSCs are able to differentiate not only into keratinocytes and fibroblasts but also into HCSs and MSCs that can also adhere to erosions [93]. In addition, the generation of autologous three-dimensional (3D) skin-equivalent grafts generated from iPSCs shows promise for the production of the stratified epidermis in vitro and in vivo [97].

At the preclinical level, several strategies using iPSCs and gene editing have been demonstrated for future personalized tissue replacement therapies. The mechanism of gene editing is based on creating double-stranded DNA breaks followed by non-homologous end joining, which can cause gene knockout or exon skipping, or homologous direct repair [45]. Using Clustered regularly interspaced palindromic repeats (CRISPR/Cas9) technology, Zinc fingers (ZFN), TALENs (transcription activator-like effector nucleases) and meganucleases in gene editing have improved the efficiency of correcting RDEB iPSCs [45]. The correction of a mutation in COLVII-A1 using CRISPR/Cas9-mediated homologous recombination with the piggyBac transposon system achieved a satisfactory genomic repair [98]. Moreover, the application of autologous skin equivalents in animal models, generated through CRISPR/Cas9, using corrected keratinocytes and fibroblasts differentiated from iPSCs was successful enough [95].

Despite the autologous origin of iPSC-derived cells, their use has still safety issues, including genetic and epigenetic instability and post-transplant efficacy and safety [99]. Therefore, extensive genetic analysis (including the whole genome) will be needed to eliminate the concern of unwanted serious complications that may occur after correcting iPSCs at the preclinical level before clinical trials can begin.

## 5. Conclusions

At present, the current evidence for the use of stem cell therapy in EB is still limited, as the overall number of patients treated in this way is low, justifying the need for further research to evaluate the effectiveness and potential risks. Data obtained from pre-clinical and clinical studies so far demonstrate that stem cells depending on the origin (autologous or heterologous), source and way of delivery (intradermal or systemic) can be beneficial and reduce disease activity in patients with RDEB. However, the mechanisms through which SCs might exert their beneficial role are still unknown or incompletely understood. The use of ex vivo gene replacement therapy through engraftment of epidermal sheets was particularly successful in the treatment of patients with JEB and RDEB. However, this therapy is only applied to the body surface to replace skin and does not address the internal complications that manifest in patients with more severe forms, such as generalized JEB. Refinement of ex vivo approaches for skin replacement and combination with stem cell infusions may be an efficient and safe solution for the treatment of EB patients in the future.

## Figures and Tables

**Figure 1 bioengineering-10-00422-f001:**
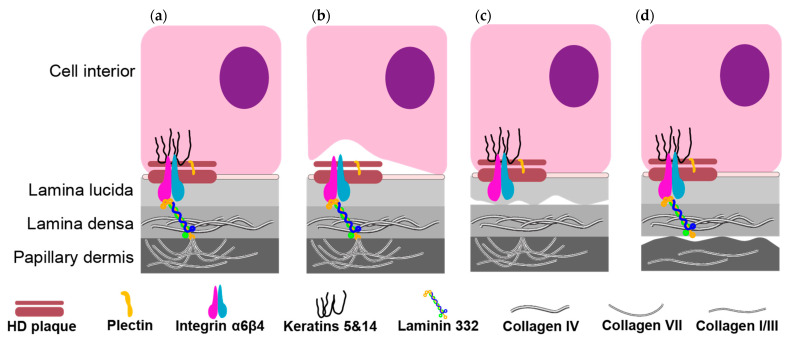
Graphic representation of the skin dermo-epidermal junction and targeted molecules associated with different types of EB. (**a**) The dermo-epidermal junction of healthy skin. The epidermal keratinocytes are strongly attached to their underlying extracellular matrix through structures called the hemidesmosomes (HD). The transmembrane receptors integrins α6β4, intermediate the connection of the extracellular laminin 332 with the keratin intermediate filaments cytoskeleton in the cell interior; (**b**) Cleavage in EBS occurs within the basal layer keratinocytes due to mutations within the genes encoding keratins, components of the intermediate filaments. The lack of the intermediate filaments cytoskeleton results in a disruption of epidermal keratinocytes strong adhesion to its underlying basement membrane at the level of the HD; (**c**) Disruption of keratinocyte cell adhesion at the level of the lamina lucida in JEB due to mutations within the genes forming laminin 332; (**d**) Cleavage at the space between the sublamina densa and in the upper papillary dermis in DEB due to absence of COLVII.

**Table 1 bioengineering-10-00422-t001:** Classification of stem cells.

Classification	Cell Type	Differentiation	Examples
Totipotent	Embryonic stem cells	Differentiate into any cell type	Zygotes
Pluripotent	Advanced Perinatal embryonic stem cells	Differentiate into of all three germ layers	ECSs, AMSCs, CMSCs, USSCs, iPSCs, VSELs
Multipotent	Adult stem cells	Differentiate into a limited range of cell types	MSCs, HSCs
Oligopotent	Adult stem cells	Differentiate into a limited range of cell types	Lymphoid, Myeloid
Unipotent	Adult stem cells	Differentiate into a single type cell type	Epidermal

**Table 2 bioengineering-10-00422-t002:** Clinical Trials using stem cell and corrected autologous cell therapies for treatment of EB (ClinicalTrials.gov).

Title/Clinical Trial Identifier	Phase/Status	Condition	Interventions/Cell Types	Safety	Outcome	Country
NCT03529877: Allogeneic ABCB5-positive Stem Cells for Treatment of Epidermolysis Bullosa [47]	Phase I/II, International, multicentric, single-arm. Completed	RDEB	Intravenous infusion of allogeneic ABCB5+ MSCs	Good tolerability, manageable safety. Adverse effects: (1) mild lymphadenopathy (1 out of 16), (2) hypersensitivity reactions (2 out of 16).	Significant reductions in EB disease activity and scarring index activity. Reductions in pain and itch.	United States (US), Austria, France, German, Italy, United Kingdom (UK)
NCT04153630: Safety Study and Preliminary Efficacy of Infusion Haploidentical Mesenchymal Stem Cells Derived From Bone Marrow	Phase I/II, pilot, single group, open-label. Unknown status	RDEB	Systemic infusion of BM-MSCs from haploidentical donor. Intravenous injection with a dose of 2–3 × 10^6^ cells/kg.	Unknown	Unknown	Spain
NCT04520022: Safety and Effectiveness Study of Allogeneic Umbilical Cord Blood-derived Mesenchymal Stem Cell [46]	Phase I/II, single-group, single-center, open-label. Completed.	RDEB	Intravenous administrations of allogeneic hUCB-MSCs, 3 × 10^6^ cells/kg, total of 3 doses every 2 weeks.	Well-tolerated, no severe adverse events.	Improvements in the EB severity score, body surface area involvement, blister counts, pain, pruritus and quality of life. Increase of collagen type VII expression at the DEJ in 1 out of 6 patients.	Korea
NCT02579369: Study to Evaluate the Safety of ALLO-ASC-DFU in the Subjects With Dystrophic Epidermolysis Bullosa	Phase I/II, non-randomized, parallel assignment, open-label. Unknown status	RDEB	Application of hydrogel dressing for RDEB wound with allogeneic AT-MSCs	Unknown	Unknown	Korea
NCT00478244: Allogeneic hematopoietic cell transplantation to correct the biochemical defect and create tolerance to donor tissue in subjects with EB [50].	Single group, open-label. Terminated (Competing studies).	RDEB	Immunomyeloablative chemotherapy and allogeneic hematopoietic stem cell transplantation	One of six patients died as a consequence of graft rejection and infection after 2 years. High-risk therapeutic approach for patients with less severe RDEB.	Increased collagen VII deposition at the DEJ in 5 of 5 recipients without normalization of anchoring fibrils and reduced blistering.	US
NCT03183934: A Follow-up Study to Evaluate the Efficacy and Safety of ALLO-ASC-DFU in ALLO-ASC-EB-101 Clinical Trial	Follow-up study		Application of dressing for DEB with allogeneic AT-MSCs	Unknown	Unknown	Korea
NCT00881556: A pilot study of reduced intensity conditioning (RIC) and allogeneic stem cell transplantation (ALLOSCT) in children with RDEB.	Early phase I, single group, open-label. Terminated	RDEB	Reduced Intensity Conditioning (RIC) and Allogeneic Stem Cell Transplantation (AlloSCT) from family-related donors and unrelated cord blood (UCB) donors will be safe and well tolerated in selected patients with RDEB.	Unknown	Unknown	U.S.
NCT02582775: Biochemical Correction of Severe EB by Allo HSCT and Serial Donor MSCs [51]	Phase II, Non-randomized, open-label, Active, not recruiting	Severe, generalized RDEB	Epidermolysis bullosa patients treated with chemotherapy and BM-HSC transplant with BM-MSCs infusions	Improved safety due to PTCγ treatment.	Improved Col VII and restoration of anchoring fibrils, reduced erosions.	U.S
NCT01033552: Biochemical Correction of Severe EB by Allo HSCT and “Off-the-shelf” MSCs	Phase I/II, single group, open-label. Completed	Severe EB	Mesenchymal stem cell transplantation infused intravenously and bone marrow or umbilical cord blood products infusion.	Unknown	Unknow	U.S.
NCT02323789: Mesenchymal allogeneic stromal cells in adults with RDEB (ADSTEM) [52]	Phase I/II, Open-label, Unknown status	RDEB	Intravenous alloegeneic mesenchymal stromal cell (BM-MSCs) therapy in adults with RDEB	No serious adverse events up to 12 months. Requirement for monitoring possible development or progression of squamous cell carcinoma (SCC).	Transient reduction in disease activity scores and significant reduction in itch. Transient increase in type VII collagen at 1 out of 10 participants.	U.K.
2012-00894-87: Allogeneic mesenchymal stromal cells for the treatment of skin disease in children with recessive dystrophic epidermolysis bullosa [53]	Phase I/II, Non-randomized-controlled, single arm. Completed	RDEB	3 intravenous allogeneic BM-MSC infusions	Good tolerance	Decrease in global severity score, increase in quality of life, decrease in blister counts.	U.K.
NCT05111600: Open-label, Pivotal Clinical Trial to Confirm Efficacy and Safety of Autologous Grafts Containing Stem Cells Genetically Modified for Epidermis Restoration in Patients With Junctional Epidermolysis Bullosa (HOLOGENE 5) [54].	Phase II/III, Prospective, multicenter and multinational, open-label, uncontrolled. Recruiting	JEB Non Herlitz type	Grafting of fibrin-cultured epidermal sheets generated by transgenic clonogenic keratinocytes, including epidermal stem cells	Promising safety profile	Most likely permanent functional restoration of the dermo-epidermal junction, long-lasting ie engrafted transgenic epidermal stem cells allowing continuous self-renewal	Italy, France
NCT02493816: Phase I Study of Lentiviral-mediated COL7A1 Gene-modified Autologous Fibroblasts in adults with Recessive Dystrophic Epidermolysis Bullosa [55].	Phase I, open-label. Completed	RDEB	Lentiviral-mediated COL7A1 gene-modified autologous fibroblasts, 3 intra-dernal injections on day 0 only	Safe, only mild local injection procedure-related side effects lasting for a few hours without requiring treatment	Significant increase in collagen VII expression at the derm-epidermla junction but not associated with mature anchoring filaments, improvement of healing.	UK
NCT02810951: A Phase I/II Study of FCX-007 (Genetically-Modified Autologous Human Dermal Fibroblasts) for Recessive Dystrophic Epidermolysis Bullosa (RDEB) [56].	Phase I/II, single group, opne-label. Terminated	RDEB	FCX-007 is a genetically modified cell product obtained from the subject’s own skin cells (Autologous fibroblasts). The cells are expanded and genetically modified to produce functional COL7. FCX-007 cell suspension is injected intradermally.			
NCT03490331: Clinical trial to assess the safety and efficacy of autologous cultured epidermal grafts containing epidermal stem cells genetically modified with a gamma-retroviral (rv) vector carrying COL17A1 cDNA for restoration of epidermis in patients with junctional epidermolysis bullosa	Phase I/II, prospective, open-label, uncontrolled. Terminated (No patient ongoing (none completed the study). Changes to the viral vector ongoing)	JEB	Transplantation surgery of genetically corrected cultured epidermal autograft.	Unknown	Unknown	Austria
NCT02984085: Clinical trial to assess the safety and efficacy of autologous cultured epidermal grafts containing epidermal stem cells genetically modified with a gamma-retroviral (rv) vector carrying COL7A1 cDNA for restoration of epidermis in patients with recessive dystrophic epidermolysis bullosa.	Phase I/II, prospective, open-label, uncontrolled. Terminated (replaced by study in progress)	RDEB	Transplantation surgery of genetically corrected cultured epidermal autograft.	Unknown	Unknown	
NCT04186650: Ex vivo gene therapy linical trial for RDEB using genetically corrected autologous skin equivalents (EBGraft).	Phase I/II, non-randomized single-group, open—label. Active, not recruiting	RDEB	Graft of SIN RV-mediated COL7A1 gene-modified autologous skin equivalent	-	-	France
NCT01263379: Gene transfer for recessive dystophic epidrmolysis bullosa [57,58,59]	Phase I/II, single-center, open-label. Active, not recruiting.	RDEB	Gene transfer for RDEB using the drug LZRSE-Col7A1 engineered autologous epidermal sheets (EB-101)	Safe, no serious adverse effects during up to 8 years. 2 adult patients out of 10 developed SCC on anatomic sites distant form grafted sites.	Long-term improvements in wound healing, pain and itch. Some grafts showed collagen VII expression in anchoring fibrils. Collagen VII expression persisted up to 2 years after treatment in 2 participants.	US
NCT04227106: A phase 3 study of EB-101 for the treatment of RDEB.	Phase III, single-group, open-label. Completed	RDEB	One-time surgical application of EB-101 on up to 6 chronic RDEB wounds.	Unknown	Unknown	US
NCT051116000: Open-label, pivotal clinical trial to confirm efficacy and safety of autologous grafts containing stem cells genetically modified for epidermis restoration in patients with JEB (HOLOGENE 5).	Phase II/III, prospective, multicenter and multinational, open-label, uncontrolled.	JEB	Transplantation of autologous cultured epidermal grafts containing epidermal stem cells genetically modified transduced with a LAMB3-gamma retroviral vector.	In progress	In progress	France, Italy.

## Data Availability

No new data were created.

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
