# Peer review of "Stem Cell Therapies for Epidermolysis Bullosa Treatment"

_bioengineering, 2023, doi:10.3390/bioengineering10040422_

Round 1

Reviewer 1 Report

The review explores potential application of stem cells in diseases of skin, the largest organ of the body with emphasis on Epidermolysis bullosa (EB) which is a group of rare skin diseases with distinct clinical characteristics and significant morbidity and mortality. The manuscript was well-written with recent trends and clinical trials as well as pros and cons, limitations of current studies. The following points must be addressed before publication.

1. The ‘1. Introduction’ section must cite Figure 1.

 2. The size of the letters in Figure 1 is too small to be seen. Use larger fonts.

 3. Inclusion of a table on different stem cells in the ‘3. Introduction to stem cells’ section may help the reader.

4. ex vivo must be italicized throughout the manuscript.

Author Response

We would like to thank the reviewer for the time taken to review this manuscript. All the comments and suggestions were fruitful and addressed from our part one by one which we consider that helped us to improve this review.

  1. The ‘1. Introduction’ section must cite Figure 1.

We thank a lot the reviewer for this comment. The Figure 1 cited on the line 71.

  1. The size of the letters in Figure 1 is too small to be seen. Use larger fonts.

We thank the reviewer for this comment and we agree that the figure was not clear enough. We have revised the entire figure and used larger fonds.

  1. Inclusion of a table on different stem cells in the ‘3. Introduction to stem cells’ section may help the reader.

We thank the reviewer for this comment. We have included A table including classification of stem cells and same examples of them cited in page 5.

  1. ex vivo must be italicized throughout the manuscript.

We thank the reviewer for this comment. We have corrected “ex vivo” in italics as is correct.

Reviewer 2 Report

The work entitled as "Stem cell therapies for epidermolysis bullosa treatment" by  Argirw Niti et al. is  very interesting and quite promising. Stems cells indeed can form a promising in vitro model for skin diseases. The work is based on scientific sound findings, is well designed and presented. I think this work is a significant contribution to this field. II endorse the publication in bioengineering.   

Author Response

We would like to thank a lot the reviewer for the time taken to revise this manuscript. We also thank the reviewer for the kind and positive comments on our work.

Reviewer 3 Report

The authors reviewed a rare skin disease, epidermolysis bullosa, whose causes come from genetic mutations of the genes involved in skin formation and its clinical trials using stem cell therapies. Although these therapies partially succeeded in improving symptoms, these are not enough to address the internal complications. Therefore, further studies should be required to develop both ex vivo gene replacement-engraftment of epidermal sheets method and genetic-modified stem cell infusions without side reactions.

This review is relatively well-structured, readable. Therefore, the reviewer thinks it deserves publication in Bioengineering.

The reviewer asks the authors for minor modifications in response to the following comments.

1. Lines 9-24, please add the current issues of stem cell therapies of EB and future perspectives as the conclusion in the abstract section.

2. Figure 1, the reviewer does not understand what the white spaces mean. Please modify the illustrations so that how the gene disruptions fail healthy skin construction. Add more detailed explanations using the images in the main text or figure legends.

3. Lines 209-215, ES cells are a part of pluripotent cells. Please merge the sentences into -the description in lines 181-184.

3. Table 1 and lines 246-460, it is hard to find the clinical trials in the table when reading sections 4.1 to 4.4. Add a number to each clinical trial and describe the number in the text part so that the reading audiences can immediately understand the trial the authors mention.

4. Please add pictures of the states of EB before and after treatment trials of stem cell therapy if possible.

Author Response

We would like to thank a lot the reviewer for the time taken to review our manuscript. All the comments and suggestions were fruitful and addressed from our part one by one which we consider that helped us to improve this review.

  1. Lines 9-24, please add the current issues of stem cell therapies of EB and future perspectives as the conclusion in the abstract section.

We thank the reviewer for this comment and we agree that conclusions were lacking from the abstract. The abstract was revised accordingly.

  1. Figure 1, the reviewer does not understand what the white spaces mean. Please modify the illustrations so that how the gene disruptions fail healthy skin construction. Add more detailed explanations using the images in the main text or figure legends.

We thank the reviewer for encouraging us to improve the quality and clearness of our figure 1. As you will be able to see in the manuscript the entire figure was re-designed and a more thorough explanation of the defected genes per EB sub-type was added in the figure legend.

  1. Lines 209-215, ES cells are a part of pluripotent cells. Please merge the sentences into -the description in lines 181-184.

We thank the reviewer for this comment. The last sentence that refers to embryonic stem cells was removed from the lines 183-184.

  1. Table 1 and lines 246-460, it is hard to find the clinical trials in the table when reading sections 4.1 to 4.4. Add a number to each clinical trial and describe the number in the text part so that the reading audiences can immediately understand the trial the authors mention.

We thank the reviewer for this comment. We added in the text the corresponding clinical trial number when applicable. However, there are some clinical trials registered in the ClinicalTrials.gov for which no results are available or that are still ongoing (so no results are available) that we did not mention in the text but we thought it was important to name them in the table.

  1. Please add pictures of the states of EB before and after treatment trials of stem cell therapy if possible.

We thank the reviewer for this comment. Unfortunately, it is not possible to provide any pictures right now.
